# Mapping Disaggregate-Level Agricultural Households in South Africa Using a Hierarchical Bayes Small Area Estimation Approach

Yegnanew A. Shiferaw

Department of Statistics, University of Johannesburg, Auckland Park Kingsway Campus, P.O. Box 524, Johannesburg 2006, South Africa; yegnanews@uj.ac.za

**Abstract:** The first important step toward ending hunger is sustainable agriculture, which is a vital component of the 2030 Agenda. In this study, auxiliary variables from the 2011 Population Census are combined with data from the 2016 Community Survey to develop and apply a hierarchical Bayes (HB) small area estimation approach for estimating the local-level households engaged in agriculture. A generalized variance function was used to reduce extreme proportions and noisy survey variances. The deviance information criterion (DIC) preferred the mixed logistic model with known sampling variance over the other two models (Fay-Herriot model and mixed log-normal model). For almost all local municipalities in South Africa, the proposed HB estimates outperform survey-based estimates in terms of root mean squared error (MSE) and coefficient of variation (CV). Indeed, information on local-level agricultural households can help governments evaluate programs that support agricultural households.

**Keywords:** agricultural households; disaggregation; hierarchical Bayes; fruits; vegetables; grains and crops





## 1. Introduction

"If we don't get sustainability right in agriculture first, it won't happen anywhere."— Wes Jackson

Agriculture, food systems, and the sustainable use of natural resources are vital to improving the livelihoods of millions of poor households in the developing world [1]. The 2030 Agenda's central theme is sustainable agriculture, which is also the first critical step toward achieving zero hunger. Despite the fact that many sustainable development goals (SDGs) touch on agricultural-related issues, SDG indicator 2.4.1 is entirely devoted to it. The SDG indicator 2.4.1 refers to the "proportion of agricultural area under productive and sustainable agriculture" [2].

Of the 1.3 billion people worldwide, 97% of them in developing countries depend highly on agriculture for employment [3]. Agriculture provides a critically important opportunity to improve the living standards of all, especially in reducing malnutrition [4]. It continues to be a primary source of food for many rural households in developing countries. Today's food and agricultural systems are incredibly productive [5]. There is enough food produced, according to academics, to meet the needs of every man and woman [6,7]. However, according to global estimates by the Food and Agriculture Organization (FAO), 2 billion people worldwide, or 26.4% of the world's population, have micronutrient deficiencies, and approximately 1.3 billion people, or 17.2% of the world's population, have experienced food insecurity at moderate levels [8]. According to [9], 653 million people would still be undernourished in 2030 under a "business as usual" scenario without additional initiatives to support pro-poor development. Note that the FAO had made this prediction before COVID-19 (a public health emergency of global concern) and the Russia–Ukraine War (the worst catastrophe to hit Europe since the Cold War).

The world witnessed a food price crisis in 2008 due to the global economic crises. The international prices were the highest in nearly 30 years during the first three months of 2008. In addition, there was social unrest in more than 40 countries between 2007 and 2009, which led to an astonishing increase in the number of hungry people [10]. The global economies were rocked by the COVID-19 outbreak in 2020 [11]. Among the affected economic sectors, the agriculture and food industries were not an exception. Food insecurity has been increasing in many countries worldwide due to the COVID-19 pandemic; the developing world has been the most affected. For instance, according to [12], COVID-19 caused an increase in food insecurity among new households by 32.3% in the first quarter of 2020. More recently, on 24 February 2022, a new armed conflict between the Russian Federation and Ukraine erupted just as the world economy was beginning to recover from the turmoil brought on by COVID-19. This war caused numerous economic repercussions, including severe food and energy crises. For example, 26 countries import 50% of their wheat needs from Russia and Ukraine, while 50 Asian and African countries rely on Russia and Ukraine for 30% of their wheat needs [13].

Like the rest of the world, South Africa was affected by the 2008–2009 economic crisis, the COVID-19 pandemic, and the Russia–Ukraine war. The prices of agricultural commodities such as fruits, vegetables, crops, and so on were high in South Africa in 2007 and 2008. There was a sharp increase in food prices in the later years. For example, the prices of food were high between 2013 and 2016 [14]. According to [15], the Russia–Ukraine conflict affects the economy through higher energy and fuel prices, which caused inflation; trade disruptions and the effects of sanctions on exports and imports; and investor uncertainty, which affected asset prices and the depreciation of the Rand (ZAR). Compared to January 2021, the price of essential agricultural inputs has increased by more than 100% in South Africa [15].

Tracking the development of agricultural programs that are sustainable and aimed at reducing hunger and malnutrition frequently requires micro or disaggregate-level estimates [16]. At the large domain level (i.e., the national or provincial level), data from household surveys typically provide a sufficiently accurate direct estimate of agricultural households. However, due to their small sample sizes, these direct estimates become less accurate as data are broken down into regional or population subgroups [16,17]. Furthermore, the distribution of agricultural households varies considerably across regions, and such large domain-level estimates hide regional differences.

When there are not enough area-specific sample observations to generate accurate direct estimates, small area estimation (SAE) approaches have been developed and are utilized to obtain estimates [18]. Any subpopulation or area of interest in SAE is referred to as an "area", which includes geographical regions (such as a state or county) and socioeconomic groupings (such as income, race, and age). SAE techniques have gained popularity over the past three decades as the focus has shifted to regional planning, creating public policies and programs, and budgetary planning at the local level instead of the provincial or national level [19]. These methods have an impact on our daily lives because they frequently act as the basis for political decision-making. They are utilized, for instance, by the World Bank to create poverty maps in developing and developed nations [16]. In order to estimate income, poverty, and the proportion of school-age children who live in poverty, the United States Census Bureau uses SAE techniques.

In South Africa, the local municipalities and wards [20] are the spatial/administrative units closest to communities because of the location of many essential services and infrastructure. In order to plan and direct efforts to reduce poverty and inequality, food insecurity, HIV prevalence, malnutrition, under-five mortality, and other issues, as well as to provide services, it is crucial to provide information at this level. Therefore, this study aims to estimate the proportion of South African households that cultivate fruits, vegetables, grains, and crops at the local municipality level using SAE techniques. Earlier research from South Africa primarily used this method to estimate a few socioeconomic indicators. For example, Ref. [21] used the income and expenditure survey 2010/11

and the Census 2011 to estimate poverty at the district and municipal levels in South Africa. In the 52 districts of South Africa, Ref. [22] showed the effectiveness of a relatively straightforward small-area estimation of HIV prevalence. Another study by [23] also uses SAE to determine the municipal-level estimates of under-five mortality in South Africa. Most recently, Ref. [24] used the Income and Expenditure Survey 2010/2011 and the 2011 national population census to estimate and map the proportion of school-age children living in poverty in South Africa's local municipalities. During COVID-19 in South African districts, Ref. [25] used SAE techniques to estimate South African resource distribution and policy impacts. As far as the author is aware, this study is the first to estimate local agricultural households in South Africa using HB models.

In this paper, three different SAE models under the Bayesian framework were developed to combine local municipality-level community survey data with local municipality-level data from auxiliary sources from the Population Census. The formulation of these SAE models in the Bayesian framework has advantages in terms of flexibility in overcoming the computational challenges of high-dimensional posterior densities integration and computational efficiency with complex small-area models. Additionally, Ref. [26] developed an R package called "BayesSAE" that is easily accessible and can be used to fit SAEs using MCMC simulations and provide model evaluation, selection, and diagnostics. Recently, the SAE approach in the Bayesian framework has been used to estimate the prevalence of having a personal doctor for all counties in the United States by combining the Behavioral Risk Factor Surveillance System survey with county-level data from auxiliary sources [27], state-level monthly unemployment rate from January 2004 to December 2007 in the United States [28], the average health expenditure for the 47 prefectures of Japan [29], and district level poverty incidence for the State of Odisha in India [30].

The disaggregated estimate of agricultural households provides critical evidence on households and provides a platform for national, regional, and provincial governments to convene the following interventions. First, the shift from subsistence to commercial agriculture is critical to economic growth and food security. Using nationally representative data from three African nations, Ref. [31] found that even the smallest and most marginalized landowners were engaging in high levels of commercialization, with market participation rates reaching 90%. The adoption of cash crops generally increased real incomes, triggering a virtuous cycle in which higher incomes were used to increase food consumption, benefiting both households in general and children in particular. Second, according to [32], local food systems have been excluded in recent years as a result of consumer preferences, globalization, and urbanization. Food systems have become industrialized, capitalistic, and owned by global behemoths. Smallholder farmers, local food manufacturers, and small and medium-sized businesses should be given more opportunities to participate in local markets without relying on large corporations. This can be accomplished by improving access to finance, strengthening the capacity of agri-food chain stakeholders, and increasing investment in local ecosystems. Third, to assist in creating new jobs and expanding the rural economy, South Africa has made considerable investments in the agricultural industry throughout the years. Although these investments have not consistently performed well, the high youth unemployment rate—among the highest in the world—is more concerning [33]. As most studies focus on national issues, Ref. [34] expresses concern about the need for more research on the spatial distribution of unemployment in South Africa and makes recommendations for the spatial analysis of unemployment, primarily in terms of currently accessible spatial and attribute data sources. Accordingly, efforts to increase youth involvement in agriculture are supported by the estimates that have been analyzed at the local level, which should be combined with strong efforts to empower young people. Finally, according to [35], the apartheid era caused disparities in the range of skills relevant to the agriculture sector in South Africa. So, by using disaggregated estimates of agricultural households (i.e., at the level of local municipalities), policymakers will be better able to address the underlying causes of any identified imbalances in the agricultural sector.

The remaining sections of the paper are organized as follows. The 2016 CS and the 2011 PC, as well as the target variables from the 2016 CS and auxiliary variables from the 2011 PC, are all described in Section 2. The model specifications also include specific linking models in the Bayesian framework as well as the well-known basic area-level Fay-Herriot model. Section 3 offers the findings, model diagnostics, and performance assessments. The results are discussed in Section 4. Finally, in Section 5, the conclusions are presented.

## 2. Materials and Methods

### 2.1. Data

South Africa is divided into nine provinces, which are further divided into district and local municipalities. Suppose $Y_{ij}$ is a binary outcome for individual $j$ ($j = 1, \cdots, N_i$) in local municipality $i$ ($i = 1, \cdots, m$) with $N_i$ the population size in local municipality $i$. Let the sampled values be denoted by $y_{ij}$, where $n_i$ is the sample size drawn from each local municipality $i$. Suppose $N = \sum_{i=1}^{m} N_i$ and $n = \sum_{i=1}^{m} n_i$ denote the total population and sample size, respectively. The focus is to estimate the true area proportions, $P_i$, in local municipality $i$, namely $P_i = \frac{1}{N_i} \sum_{j=1}^{N_i} Y_{ij}$ that in this case, can be estimated from the CS data using the direct survey estimator as follows $p_{iw} = \frac{1}{\sum_{j=1}^{n_i} w_{ij}} \sum_{j=1}^{n_i} w_{ij} y_{ij}$, where $w_{ij}$ represents the CS weight. Using information from the 2016 CS and the 2011 PC, we applied the SAE approach to derive estimates for local municipalities of the proportion of households engaged in growing fruits, vegetables, grains, and food crops. For this analysis, the following variables were necessary.

The 2016 CS data from Statistics South Africa generated the **target variables** . The main goal of this survey was to collect information on household size, estimated population size, fertility, mortality, migration, employment, unemployment, and the degree of household poverty. The CS data was obtained from the Stats SA website, which can be found at http://nesstar.statssa.gov.za:8282/webview/; accessed on 13 November 2019. In the 2016 CS, $m = 214$ local municipalities were sampled. The proportion of households growing fruits and vegetables, as well as the proportion of households growing grains and food crops, are the variables of interest at the local municipality level. **Auxiliary aggregate data** from the 2011 South African Census were used in addition to the 2016 CS data. The following covariates, unless otherwise stated, were evaluated at the local municipality level during model selection: sex (% female, % male); age in years (% 0–14, % 15–24, % 25–34, % 35–44, % 45–54, % 55–64 and % $\geq$ 65 years); race (% Colored, % Black African, % White and % Indian or Asian); status of employment (% employed, % unemployed, and % economically inactive); status of employment of the head (% employed, % unemployed and % economically inactive); sectors of employment (% private households, % the informal sector and % the formal sector); marital status (% married, % living together as a married couple, % never married, % widower/widow, % separated, % divorced); education level (% no formal education, % some form of primary education, % some form of secondary education, % grade 12, % tertiary education, and % other); % urban area, % farm area, income in ZAR (% no source of income, % R 1–R 76 800, % R 76 801–R 614 400, and % R 614 401 or more). Because the proportions of people in the auxiliary variable categories added up to one, the reference categories were dropped from the analysis [36,37]. The DataFirst website was used to access the 2011 South African PC data: https://www.datafirst.uct.ac.za/dataportal/index.php/catalog/485 ; accessed on 13 November 2019.

Following correlation analysis, pertinent covariates were selected from these datasets using step-wise regression analysis [38]. In order to determine whether one of the covariates—for instance, the employed category—correlates reasonably well with the proportion of households producing fruit, a correlation analysis was first carried out. Each target and auxiliary variable underwent a second evaluation. Finally, the following variables were selected for further investigation (Table 1):

**Table 1.** Variables chosen for further analysis.

| Response Variables | Covariates |
|---|---|
| Proportion of households<br><br>engaged in producing fruits | % 25–34, % no formal education, % some form of primary education, % some form of secondary education, % employed, % economically inactive, % head employed, % head unemployed, % colored, % married, % never married, % R 1–R 76, 800, |
| Proportion of households engaged in producing vegetables | % 25–34, % 35–44, % no formal education, % some form of primary education, % some form of secondary education, % employed, % economically inactive, % head employed, % head economically inactive, % White, % the formal sector, % No income, % R 76, 801–R 614, 400 |
| Proportion of households engaged in producing grains and food crops | % 35–44, % 45–54, % female, % no formal education, % employed, % economically inactive, % head employed, % head economically inactive, % never married, % divorced, % Black African, % White, % No source of income, % R 1–R 76, 800 |

### 2.2. Small Area Estimation Methodology

SAE techniques frequently use area (or aggregate)-level models when unit-level data are unavailable due to confidentiality issues or when auxiliary variables are only available in aggregate form. SAE uses data from various sources to estimate results for small geographic areas or subpopulations [18]. Numerous other small area works of literature and methodological advancements have occurred since Fay and Herriot's seminal work on the area-level small area model (often referred to as the Fay-Herriot (FH) model) in 1979.

#### 2.2.1. The Fay-Herriot (FH) Model

Basic area-level FH models incorporate auxiliary variables derived from secondary sources, such as census or administrative records, along with direct aggregate (local municipality) level survey estimates. Consider the area-level data $\{p_{iw}, \mathbf{x}_i, i = 1, \cdots, m\}$, where $p_{iw}$ is the direct survey-based estimator of $P_i$ with $E(p_{iw}|P_i) = P_i$ and $\text{Var}(p_{iw}|P_i) = D_i$, $\mathbf{x}_i$ is a $p \times 1$ area-specific covariates, often obtained from administrative records or population census. It is assumed that the sampling variances, $D_i$, are known. The sampling variances, $D_i$, are assumed to be known. The FH model (see [39]) also known as the basic area-level model composed of two parts. The natural sampling model is described in the first section as follows. It accounts for the sampling variability of direct survey estimates ($p_{iw}$) of the population parameter $P_i$, which is given by

$$p_{iw} = P_i + \epsilon_i, \quad i = 1, \cdots, m, \tag{1}$$

where $P_i$ denotes the true small area proportion that is to be estimated, $\epsilon_i \sim N(0, D_i)$ are independent and normally distributed sampling error associated with $p_{iw}$.

The linking model, the second component, connects the population parameter $P_i$ to a vector of well-known auxiliary variables ($\mathbf{x}_i$). It can be expressed as follows:

$$P_i = \mathbf{x}_i^T \boldsymbol{\beta} + v_i; \ i = 1, \cdots, m \tag{2}$$

where $\boldsymbol{\beta} = (\beta_1, \cdots, \beta_p)^T$ is a $p \times 1$ unknown regression coefficient and $v_i \sim N(0, \sigma_v^2)$ are independent and identically distributed area-specific random effect that account for between areas dissimilarities. It is assumed that the distributions of $v_i$ and $\epsilon_i$ are independent. It is common to assume that two random error components follow a normal distribution when estimating the target parameter and the MSE of the estimate. When the aim of inference is proportion, however, assuming a linear linkage model with normal random effects may

lead to incorrect and erroneous estimates [18]. According to [30], the logistic or logit link function is best for solving the issue. As a result, the linking model is expressed as:

$$logit(P_i) = \mathbf{x}_i^T \boldsymbol{\beta} + v_i, \ i = 1, \cdots, m,$$ (3)

with

$$P_i = \frac{e^{\mathbf{x}_i' \boldsymbol{\beta} + v_i}}{1 + e^{\mathbf{x}_i' \boldsymbol{\beta} + v_i}}.$$

### 2.2.2. The Hierarchical Bayes (HB) Approach

The HB approach is used in this paper to estimate the small area proportions [30]. In this approach, the HB predictors of $P_i$ are obtained by assuming some prior distribution for unknown parameters such as $\beta$ and $\sigma_v^2$. The HB inference is based on the posterior distribution of $P_i$ given all the sample observations in all the areas. Under squared error loss, the parameters of interest, $P_i$, are estimated by the posterior mean and its uncertainty or error is measured by the posterior variance. Because the HB approach can successfully handle complex small-area models using Monte Carlo Markov Chain (MCMC), it avoids the computational challenges of high-dimensional integration of posterior densities. An HB approach for estimating the small area parameters was proposed by [40] for a specified linking model.

Following [30], we propose three HB models as shown below.

**Model 1 (M1)**: The FH model (Equations (1) and (2)) with known sampling variance

Sampling model: $p_{iw} = P_i + \epsilon_i, \ i = 1, \cdots, m.$
Linking model: $P_i = \mathbf{x}_i^T \boldsymbol{\beta} + v_i, \ i = 1, \cdots, m.$

**Model 2 (M2)**: The two-stage mixed log-normal model, which incorporates both the sampling and linking models, is given as follows:

Sampling model: $p_{iw} = P_i + \epsilon_i, \ i = 1, \cdots, m.$
Unmatched linking model: $log(P_i) = \mathbf{x}_i^T \boldsymbol{\beta} + v_i, \ i = 1, \cdots, m.$

**Model 3 (M3)**: The two-stage mixed logistic model with known sampling variance, which includes both the linking model and the sampling model, is given as follows:

Sampling model: $p_{iw} = P_i + \epsilon_i, \ i = 1, \cdots, m.$
Unmatched linking model: $logit(P_i) = \mathbf{x}_i^T \boldsymbol{\beta} + v_i, \ i = 1, \cdots, m.$

Both M2 and M3 are unmatched area-level models [18].

### Prior Specification for the Parameters in the Model

The application of the HB method under the above models consists of the following important steps [41]:

(i) Specify prior distributions for the model parameters $\boldsymbol{\beta}$ and $\sigma_v^2$. The Bayesian inference drawn from posterior distribution is highly dependent on a wide range of prior distributions. Therefore, prior specification for the parameters in the model plays a vital role in Bayesian inference [30]. Following [30], we considered uniform prior for the prior variance $\sigma_v^2$, that is Uniform $(0, 10^3)$. Furthermore, for $\boldsymbol{\beta}$ prior, we considered a normal distribution, that is $N(0, 10^6)$.

(ii) Generate data from the posterior distributions of $\boldsymbol{\beta}$, $\sigma_v^2$ and $v_1, \cdots, v_m$ from MCMC simulations. The HB estimates of the true proportions $\theta_i$ obtained from each of the priors were approximated by 50,000 independent samples of the MCMC method.

Lastly, we used the deviance information criterion (DIC, [42]) to choose the best candidate model. The smaller the DIC values, the better the model. A Bayesian credible interval was also used to quantify uncertainty. One of the most widely used measures of uncertainty in Bayesian inference is the Bayesian credible interval, comparable to the confidence interval in the frequentist approach. The lower limit of the 95% credible equal tail interval is the quantile, which represents a probability of 0.025 (or the 2.5 percentile of the posterior distribution). Comparatively, the quantile, or the 97.5 percentile of the posterior distribution, represents the upper bound of the equal-tail credible interval and has a probability of 0.975 [43].

2.2.3. Generalized Variance Function

Most of the time, it was assumed that the sampling variance, $D_i$, was known. In specific applications, this supposition might need to be revised. This known sampling variance assumption can be relaxed by estimating $D_i$ from 2016 CS data and smoothing the estimated variances to obtain a more stable estimate of $D_i$ [18]. It is common practice in large-scale sample surveys to approximate the design variance of point estimators for population means and proportions using generalized variance function (GVF) models, a data analysis technique [44]. In order to obtain smoothed estimates of the sampling variances, this model was used [45]. The bias-corrected GVF was taken into account for this study by fitting the simple linear regression model given in [37]:

$$\log(\text{var}(p_{iw})) = b_0 + b_1 p_{iw} + \epsilon_i, \tag{4}$$

where $\log(\text{var}(p_{iw}))$ is the dependent variable, $p_{iw}$ as independent variable, $b_0$ and $b_1$ are the least square estimates and $\epsilon_i \overset{iid}{\sim} N(0, \sigma^2)$, $i = 1, ..., m$. Then, the GVF motivated from the above model was given by:

$$\widehat{\text{GVF}}_i = \exp(\frac{\hat{\sigma}^2}{2})\exp(\hat{b}_0 + \hat{b}_1 p_{iw}) \tag{5}$$

the factors $\left( \exp(\frac{\hat{\sigma}_{Fruits}^2}{2}) = 1.916, \ \exp(\frac{\hat{\sigma}_{Vegetables}^2}{2}) = 2.492, \ \exp(\frac{\hat{\sigma}_{Grains}^2}{2}) = 2.401 \right)$ is the bias-correction terms in the log-linear analysis for the proportion of households engaged in producing fruits, the proportion of households engaged in vegetables and the proportion of households engaged in grains and food crops, respectively. The true variance will be underestimated when we fail to consider the correction term in the GVF approach, as described in [37]. In Figure 1, direct survey estimates of the proportion of households growing fruits, vegetables, grains, and food crops are compared to sampling variance ($D_i$) and predicted variance ($\widehat{\text{GVF}}$) estimates. It is clear from these plots that the GVF eliminates the noisy and unreliable estimated variance [18,37].

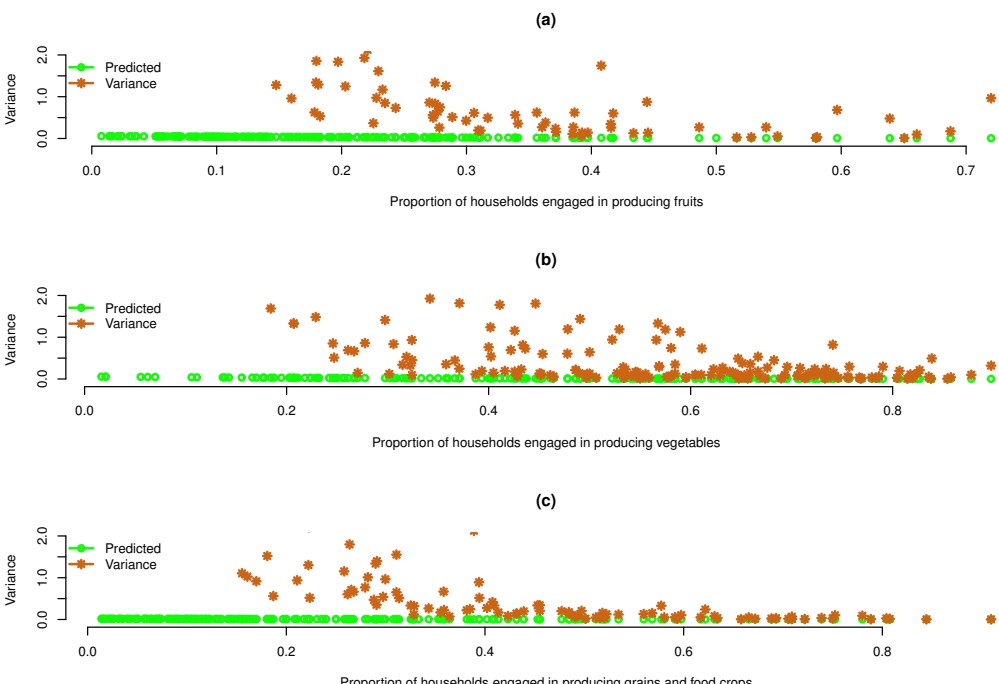

**Figure 1.** The dispersion plots for GVF fit.

### 3. Results

The proportions of households producing fruits, vegetables, grains, and food crops were used as the response variables, and the covariates shown in Table 1 were used as auxiliary variables to fit three different models (the FH model, the mixed log-normal model, and the mixed logistic model). Table 2 displays the DIC values for the suggested models. As indicated by the light blue color, the goodness-of-fit statistics preferred the mixed logistic model with known sampling variance.

**Table 2.** The values of the DIC for the three models.

| Variable Categories | M1 | M2 | M3 |
|---|---|---|---|
| Fruits | 158.26 | 164.57 | 146.38 |
| Vegetables | 745.02 | 728.47 | 726.19 |
| Grains and food crops | 141.93 | 133.63 | 128.23 |

Table 3 shows the posterior mean, standard deviations, and 95% credible intervals for the mixed logistic model with HB inference for fruits. The results of this study showed that the head unemployed, intercept, and not economically active were all highly significant predictors of 'the proportion of households producing fruits.' None of the age groups, educational levels, racial groups, marital statuses, or income statuses significantly impacted the proportion of 'households producing fruits'.

Table 4 displays the posterior mean, standard deviations, and 95% credible intervals for the mixed logistic model with HB inference for vegetables. The factors that significantly influenced the 'proportion of households producing vegetables' included not having attended school, having completed grade 12, being white, working in the formal sector, and having an income between 76,801 and 614,400. The proportion of 'households producing fruits' was not significantly influenced by any of the age groups, employment conditions, or marital statuses.

**Table 3.** The posterior means with standard deviations and 95% credible intervals for the model parameters of the mixed logistic model with HB inference: fruits.

| Covariate | Posterior Mean | Std.Dev | Credible Interval Lower | Upper |
|---|---|---|---|---|
| Intercept | −19.53 | 9.45 | −37.94 | −1.47 |
| Age (years) | | | | |
| 25–34 | −3.74 | 7.55 | −18.32 | 11.06 |
| Education | | | | |
| no schooling | −3.07 | 3.42 | −9.81 | 3.63 |
| some primary | −5.39 | 3.38 | −12.07 | 1.21 |
| some secondary | 1.29 | 3.63 | −5.86 | 8.44 |
| Employment | | | | |
| employed | 15.13 | 8.06 | −0.47 | 30.92 |
| not economically active | 15.6 | 7.85 | 0.28 | 30.78 |
| head employed | 3.18 | 5.27 | −6.96 | 13.67 |
| head unemployed | 27.16 | 10.92 | 5.92 | 48.65 |
| Race | | | | |
| colored | 0.01 | 0.61 | −1.19 | 1.22 |
| Marital status | | | | |
| married | −0.52 | 4.02 | −8.22 | 7.41 |
| never married | −0.53 | 3.13 | −6.65 | 5.52 |
| Income (South African Rand) | | | | |
| 1–76,800 | 3.6 | 2.32 | −0.93 | 8.18 |
| Variance of residual | | | | |
| | 0.38 | 0.11 | 0.21 | 0.62 |

**Table 4.** The posterior mean with standard deviations and 95% credible intervals for the model parameters of the mixed logistic model with HB inference: vegetables.

| Covariate | Posterior Mean | Std.Dev | Credible Interval Lower | Upper |
|---|---|---|---|---|
| Intercept | −0.44094 | 2.861044 | −6.09976 | 5.08 |
| Age (years) | | | | |
| 25–34 | 2.342533 | 4.727152 | −6.81574 | 11.756 |
| 35–44 | −7.41626 | 8.006608 | −23.0478 | 8.332 |
| Education | | | | |
| no schooling | −5.89356 | 1.788962 | −9.48592 | −2.4 |
| some primary | −0.85429 | 2.65225 | −6.01582 | 4.305 |
| some secondary | 1.437758 | 7.287325 | −12.9459 | 15.414 |
| grade 12 | 7.989928 | 2.512446 | 3.107896 | 13.001 |
| Employment | | | | |
| employed | 7.576255 | 5.434107 | −2.93134 | 18.364 |
| not economically active | 9.938161 | 5.153727 | −0.00857 | 20.21 |
| head employed | −7.44145 | 7.579633 | −22.4186 | 7.361 |
| head not economically active | −10.9678 | 6.885108 | −24.6622 | 2.373 |
| Race | | | | |
| white | 5.391023 | 1.784264 | 1.957281 | 8.931 |
| Employment sector | | | | |
| formal sector | 2.091212 | 0.880927 | 0.395146 | 3.838 |
| Income (South African Rand) | | | | |
| no income | 2.266863 | 2.997185 | −3.46953 | 8.109 |
| 76,801–614,400 | −8.06922 | 1.752026 | −11.6278 | −4.685 |
| Variance of residual | | | | |
| | 0.29 | 0.06 | 0.19 | 0.41 |

The posterior mean, standard deviations, and 95% credible intervals for the mixed logistic model with HB inference are also shown in Table 5 for households growing grains and food crops. Similarly, 'the proportion of households growing grains and food crops' was significantly influenced by having no formal education, being employed, being the head of a household employed, and being white. None of the age groups, income levels, or marital statuses significantly affected the proportion of 'households growing grains and food crops'.

**Table 5.** The posterior mean with standard deviations and 95% credible intervals for the model parameters of the mixed logistic model with HB inference: grains and food crops.

| Covariate | Posterior Mean | Std.Dev | Credible Interval Lower | Credible Interval Upper |
|---|---|---|---|---|
| Intercept | 25.511 | 20.867 | −3.544 | 69.304 |
| Age (years) | | | | |
| 35–44 | 40.359 | 35.598 | −21.468 | 120.32 |
| 45–54 | −15.002 | 35.711 | −102.642 | 37.476 |
| Sex | | | | |
| female | 17.112 | 14.723 | −11.302 | 42.296 |
| Education | | | | |
| no schooling | −12.517 | 8.279 | −29.596 | −0.151 |
| Employment | | | | |
| employed | 37.911 | 18.536 | 2.58 | 76.272 |
| not economically active | 7.568 | 20.515 | −27.01 | 47.972 |
| head employed | −71.493 | 29.735 | −134.078 | −21.611 |
| head not economically active | −32.872 | 27.627 | −93.574 | 12.65 |
| Marital status | | | | |
| never married | 2.513 | 5.138 | −7.358 | 12.107 |
| divorced | −125.488 | 93.014 | −325.381 | 28.285 |
| Race | | | | |
| black | 1.045 | 1.27 | −1.731 | 3.5 |
| white | 20.505 | 6.625 | 5.566 | 30.396 |
| Income (South African Rand) | | | | |
| no income | −22.385 | 17.068 | −55.903 | 4.73 |
| 1–76,800 | −10.479 | 8.486 | −24.929 | 3.852 |
| Variance of residual | | | | |
| | 0.73 | 0.31 | 0.35 | 1.58 |

Figure 2 shows the root mean squared errors (MSEs) of the HB and direct estimators. The root MSE of the HB estimator is, as can be seen in the figure, lower than the root MSE of the direct survey estimators. The distribution of the coefficient of variation (CV) for the direct and HB estimates is shown in Figure 3. This figure demonstrates that the estimated CVs for HB estimates are more reliable than those for direct survey-based estimates.

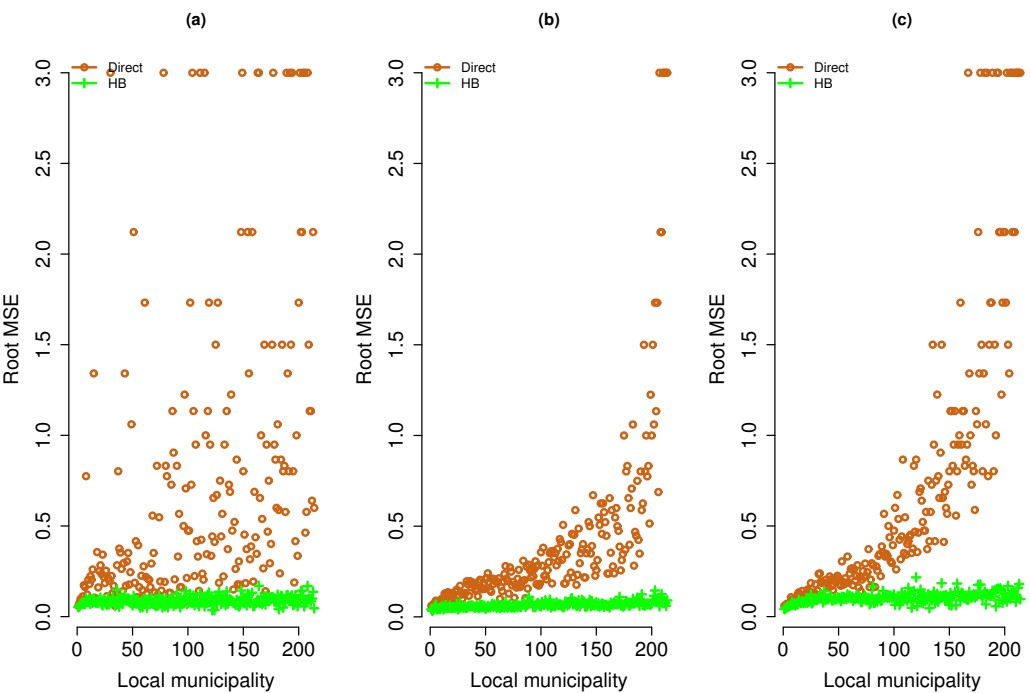

**Figure 2.** The root MSEs of the direct and HB estimates of agricultural households for each South African local municipality: Fruits (**a**); vegetables (**b**); and grain and food crops (**c**).

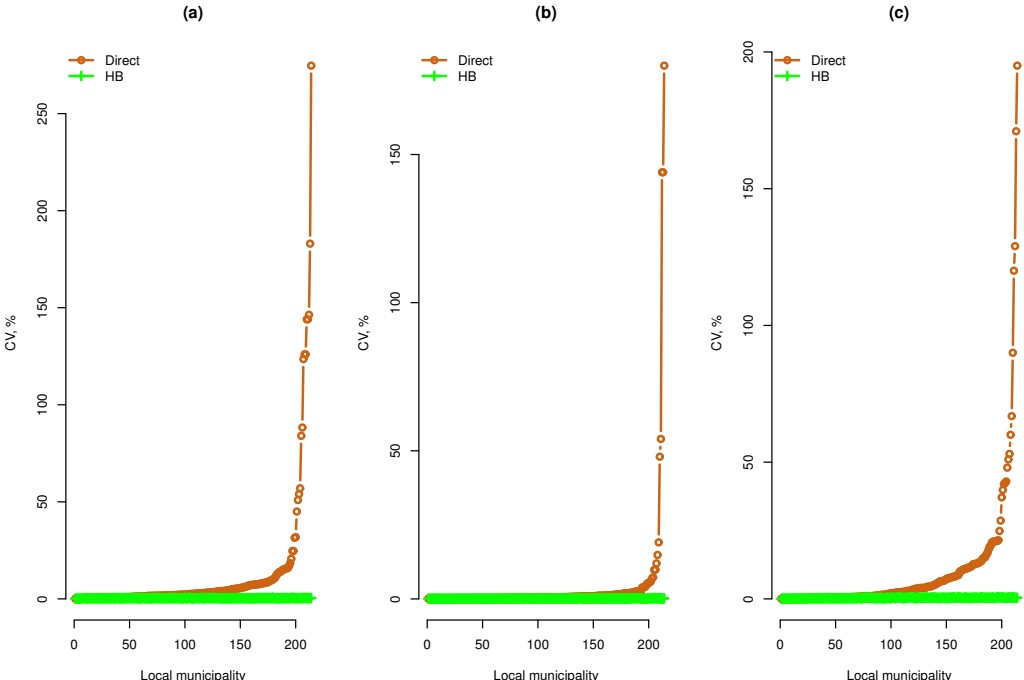

**Figure 3.** The CV of direct and model-based estimates for: (**a**) the share of households producing fruits; (**b**) the share of households producing vegetables; and (**c**) the share of households producing grains and food crops.

The histogram of residuals (left) and standard normal q-q plots of residuals (right) are shown in Figure 4. These plots support the assumption that the residuals are normally distributed. The right plots demonstrate that the q-q plots of these residuals are uniformly distributed along a straight line. These plots offer proof in favor of the residuals' normality

assumption. Additionally, the normality assumption of the standardized residuals was evaluated using the Shapiro–Wilk normality test. The Shapiro–Wilk normality test on the residuals in this research yielded *p*-values for fruits, vegetables, grains, and food crops of 0.2982, 0.5828, and 0.2162, respectively, indicating the non-rejection of the hypotheses of normality.

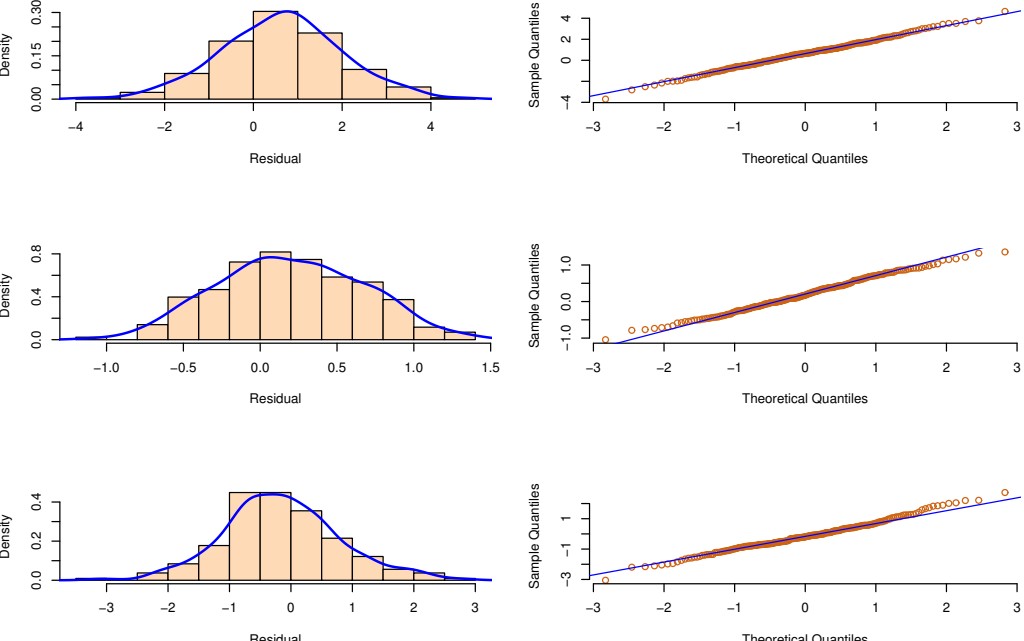

**Figure 4.** Histogram of residuals along with normal fit (**left**) and q-q plots for the standardized residuals (**right**) for the proportion of households engaged in producing fruits (**top**), the proportion of households engaged in producing vegetables (**middle**) and the proportion of households engaged in producing grains and food crops (**bottom**).

Finally, we used [46]'s goodness-of-fit diagnostic to determine whether the direct survey and model-based estimates are statistically equivalent. The null hypothesis of interest is as follows: $H_0$: the direct and HB estimates are statistically equivalent. The Wald statistic for the proposed model's goodness-of-fit diagnostic is as follows:

$$W = \sum_i \left[ \frac{(p_{iw} - \hat{P}_i^{HB})^2}{\widehat{\text{GVF}}_i + \hat{V}(P_i|\hat{P})} \right] \sim \chi^2(m), \tag{6}$$

where $\hat{V}(P_i|\hat{P})$ denotes the posterior variance estimate of $P_i$ and $\widehat{\text{GVF}}_i$ denotes the GVF of $p_{iw}$. In our case, the p-values for the proportion of households producing fruits, vegetables, and grain and crop production are 0.150, 0.186, and 0.115, respectively. These values are greater than 0.05, indicating that HB estimates are consistent with direct estimates.

Finally, Figure 5 depicts the spatial mapping of local municipality-level estimates of the proportion of households engaged in fruit production, the proportion of households engaged in vegetable production, and the proportion of households engaged in grain and food crop production.

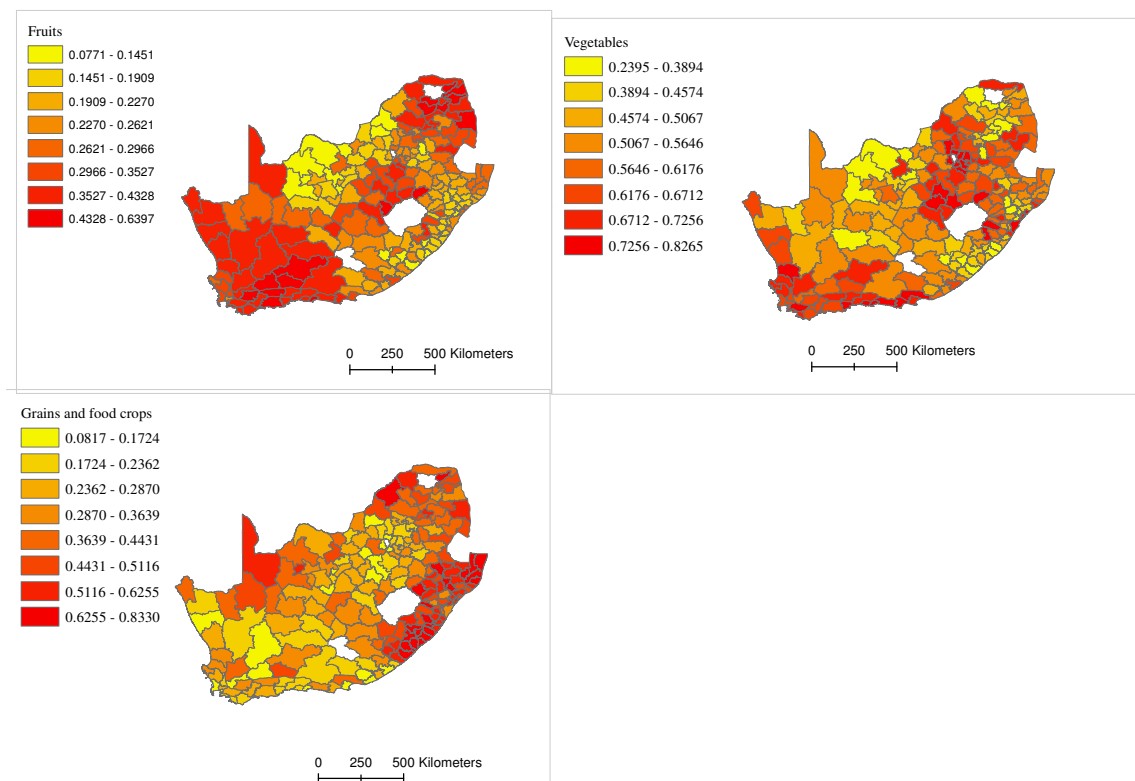

**Figure 5.** Estimates of the households engaged in agriculture by the local municipality in South Africa using the 2016 CS and the 2011 PC. The obtained estimates are the proportions of households producing fruits (**top left**), vegetables (**top right**), grains, and food crops (**bottom left**) under the model-based approach (i.e., HB approach).

## 4. Discussion

In the past two decades, as the focus shifted from provincial and federal policies to local health interventions, SAE techniques have grown in popularity [18,19]. Three dependent variables were used in this study: the proportion of households growing grains and food crops, the proportion of households growing vegetables, and the proportion of households growing fruits. Complex models are used for SAE when responses take the form of counts or proportions. Due to this reason, Bayesian methods are used in this research because they are computationally more practical than the maximum likelihood approach [26,30]. This study combined the 2016 CS and 2011 PC datasets to estimate the proportion of local municipality-level households using SAE modeling methods within the Bayesian framework. The approach provides the South African government and stakeholders with a low-cost, cutting-edge monitoring system for agricultural households.

The HB estimates should be consistent with unbiased survey-based estimates and more precise than survey-based estimates. For this purpose, we used a variety of model diagnostics, including residual analysis (histogram of residuals, q-q plot of the standardized residuals, the Shapiro–Wilk normality test), the Wald statistic, the CVs, and root MSEs, to assess the reliability and validity of the model-based small area estimates (also known as HB estimates). Because the random effects in the FH model are independent, identical, and normally distributed with a mean of zero, it is vital to validate the model assumptions; for more information, see [18]. Additionally, the optimality properties of the model-based estimates depend on the extent to which distributional assumptions are valid [18]. The histograms, the q-q plots, and the Shapiro–Wilk test all support the normality of the standardized residuals. Furthermore, Ref. [46]'s goodness-of-fit diagnostic revealed that HB estimates are consistent with direct estimates. In general, the root MSE of the corresponding HB estimators was lower than the root MSE of the direct survey estimators (Figure 2). This

is due to one of the characteristics of the class of model-based unbiased linear predictors, which is their capacity to minimize the model-based MSE [18,47]. Furthermore, HB estimators had lower CVs than corresponding direct estimators (Figure 3). The HB estimators' improvement in precision tended to be more pronounced in the local municipalities on the right side of the plot (local municipalities with small sample sizes). In other words, we significantly improved the HB local municipality level estimates by borrowing strength from the 2011 PC, especially in the municipalities with sparse samples. This fact was found using FH and spatial FH models by [18,23,38].

We found that estimates of agricultural households differ significantly across South African local municipalities using the HB method in the SAE framework (see Figure 5). In general, the maps show that local municipalities with the highest proportion of agricultural households producing fruits are found in the Western Cape, Northern Cape, Free State, and Limpopo provinces. The Western Cape, the Free State, and portions of Gauteng are home to the local municipalities with the highest concentration of households that grow vegetables. Most households that cultivate grains and food crops reside in local municipalities in the Northern Cape, KwaZulu Natal, and Limpopo provinces. To be more precise, the percentage of households producing fruits varies by the local municipality and ranges from 7.71% to 63.97%. These spatial differences would go unnoticed if only national or provincial estimates were used, allowing agricultural policies and interventions to be focused on particular local areas. The disaggregated estimates produced by this research may have an impact due to the possibility that each local government has a different set of objectives. This is so that municipalities, subject to the national and provincial laws outlined in Chapter 7 of the South African Constitution, Section 151 (3), have the authority to manage the local government affairs of their communities independently. Additionally, agricultural agencies may consider initiatives that, among other things, provide digital agricultural solutions [48], deliver agricultural extension services, and encourage the adoption of cash crops [31,32] in light of the fact that this research identified local municipalities with larger agricultural households.

The 95% credible intervals for covariate coefficients, such as head unemployed and not being economically active, do not include zero (Table 3), implying that the associated auxiliary variables account for a significant portion of households producing fruits. Similarly, the 95% credible intervals for covariate coefficients, such as not having attended school, having completed grade 12, and being white, do not include zero (Table 4), implying that the associated auxiliary variables account for a significant portion of households producing vegetables. Additionally, the 95% credible intervals for the covariate coefficients—such as having no formal education, being employed, being the head of a household employed, and being white, do not include zero (Table 5), suggesting that the associated auxiliary variables account for a sizeable portion of households growing grains and food crops.

## 5. Conclusions

Disaggregated data on population and development indicators that enable mapping of demographic inequalities and social and economic disparities are becoming increasingly crucial for programmatic and policy purposes; however, direct estimation for small areas requires more than population-based surveys [49]. In other words, for many important indicators, such data are not available at a higher resolution. In this study, the SAE method is used to produce accurate estimates and maps demonstrating how disaggregated households producing fruits, vegetables, grains, and crops can be tracked at a high level of granularity and at a reasonable cost. Although SAE applications are computationally intensive, they have a low economic cost because they do not necessitate extensive data collection efforts. The method uses open-source R software to combine information from the 2016 CS and the 2011 PC. The results show that the local municipality-level estimates of agricultural households produced by the HB method under the MCMC are more precise when applied to auxiliary data from the 2011 PC.

To see the variations between municipalities, we use geospatial maps of agricultural households (Figure 5). The municipalities with a high proportion of agricultural households (hot spots) in 2016 indicate the localized areas with the highest proportion of farms (especially those producing fruits, vegetables, grains, and food crops). Additionally, the Department of Agriculture, Land Reform, and Rural Development can use our HB estimates to accelerate land reform, improve agricultural production, better understand regional variations in agricultural households, and develop other effective policies (at the lower levels of geography, i.e., local municipalities).

Nowadays, it is essential to have access to comprehensive, high-quality data on businesses at the local level to assess the effectiveness and competitiveness of modern economic systems and to make it simpler to develop policy measures that ensure productivity and employment growth [50,51], especially in the agricultural industry. The availability of local municipality-level agricultural household statistics in this research is essential for monitoring policy and planning because local municipalities are a crucial component of the national planning process. Additionally, our data-driven HB estimates at the local municipality level will be able to help decentralized planning improve the efficacy of development programs, particularly SDG indicator 2.4.1 [2].

Some limitations are associated with this study. As described in the previous sections, this study extracts key auxiliary variables from the 2011 PC, which was the most recent population census available at the time of our analysis. It combines them with the 2016 CS to produce more precise local municipality-level estimates of agricultural households in South Africa. These datasets cover different time periods, which is one of the limitations of this study. Future research should use data from the 2022 Population Census and compare it to previous studies to identify variations in agricultural households. Given that the most recent census was conducted in 2022, and the report is most likely to be made public in 2024, we will have the most recent data on local-level agricultural household statistics at our disposal. After assessing the information from the Census 2022 dataset, the best HB model will be applied case-by-case for other agriculture-related variables.

**Funding:** This research received no external funding.

**Institutional Review Board Statement:** Not applicable.

**Data Availability Statement:** The 2011 PC can be accessed at http://nesstar.statssa.gov.za:8282/webview/ accessed on 13 November 2019, and the 2016 CS can be accessed at https://www.datafirst.uct.ac.za/dataportal/index.php/catalog/485 accessed on 13 November 2019.

**Acknowledgments:** For the CS2016 and 2011 Census datasets, the author is grateful to Statistics South Africa and Data First.

**Conflicts of Interest:** The author declares no conflict of interest.

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
