# Peer review of "Mapping Disaggregate-Level Agricultural Households in South Africa Using a Hierarchical Bayes Small Area Estimation Approach"

_agriculture, doi:10.3390/agriculture13030631_

Round 1

Reviewer 1 Report

 The manuscript draft is devoted to an interesting problem that touches on estimating the local-level households engaged in agriculture. The authors build the HB model to estimate local municipality-level households in South Africa that produce fruits, vegetables, grains, and crops. The proposed approach is logical, results are clear. However, I have the following remarks:

1.      The title is too long. The authors should shorten the title of the paper.

2.      Introduction section is good. In the final part of the introduction, a brief overview of the rest of the paper should be written. It is appropriate for the authors to explain not only the structure but also the logic of the remainder of the paper.

3.      The reference needs improvement. The geography of cited publications should also be widened.

4.      The results are not explained properly. The discussion section should be extended using: a discussion of related research, and a comparison between the authors’ results and initial hypothesis.

5.      It would be of utmost importance that the authors provide a sound theoretical framework for your case study, explain the objectives and contributions of their research, and ensure that they adequately justify how the finding of agricultural households can be constructed based on the data from 2011 and 2016.

6.      The authors should show the feasibility of using their results in practice.

7.      In the conclusions section, the authors should summarize the paper. The conclusion should also emphasize the limitations of the author's approach. It would be better to discuss the implications for a business. The conclusion section should be extended using: a discussion of related research, and a comparison between the authors’ results and initial hypothesis and implications for future research.

Author Response

Reviewer #1's comments are addressed point-by-point in the attached document. Please see the attachment.  

Reviewer 2 Report

The manuscript addresses important and current issues related to mapping farm households in South Africa using a hierarchical Bayes small area estimation approach by linking big data sources.

Please consider whether it is necessary to refer to the same literature source more than once in the same paragraph, e.g. "First, the shift from subsistence to commercial agriculture is critical to economic growth and food security. Using nationally representative data from three African nations, [21] found that even the smallest and most marginalized landowners were engaging in high levels of commercialization, with market participation rates reaching 90%. Adoption of cash crops generally increased real incomes, triggering a virtuous cycle in which higher incomes were used to increase food consumption, benefiting both households in general and children in particular [21]. Second, according to [22], local food systems have been excluded in recent years as a result of consumer preferences, globalization, and urbanization. Food systems have become industrialized, capitalistic, and owned by global behemoths. Smallholder farmers, local food manufacturers, and small and medium-sized businesses should be given more opportunities to participate in local markets without relying on large corporations [22]."

The Conclusions chapter needs improvement. Please underline the importance of the analyzes performed in the manuscript in the Conclusions chapter. Please try to indicate the possible application of the research results and highlight what the author of the manuscript brought to science. Please also try to identify directions for future research related to the topic of the manuscript.

Author Response

Reviewer #2's comments are addressed point-by-point in the attached document. Please see the attachment.

Round 2

Reviewer 1 Report

The paper can be accepted in its present form